# Actin-Dependent Mechanism of Tumor Progression Induced by a Dysfunction of p53 Tumor Suppressor

**DOI:** 10.3390/cancers16061123

**Published:** 2024-03-11

**Authors:** Natalia Khromova, Maria Vasileva, Vera Dugina, Dmitry Kudlay, Peter Chumakov, Sergei Boichuk, Pavel Kopnin

**Affiliations:** 1Carcinogenesis Institute, N.N. Blokhin National Medical Research Oncology Center, The Ministry of Health of Russia, 115478 Moscow, Russia; nkhromova@gmail.com (N.K.); mvnovikova94@mail.ru (M.V.); vdugina@iname.com (V.D.); 2Belozersky Research Institute of Physico-Chemical Biology, Lomonosov Moscow State University, 119992 Moscow, Russia; 3Biological Faculty, Lomonosov Moscow State University, 119991 Moscow, Russia; 4Department of Pharmacology, The I.M. Sechenov First Moscow State Medical University (The Sechenov University), 119991 Moscow, Russia; d624254@gmail.com; 5Department of Pharmacognosy and Industrial Pharmacy, Lomonosov Moscow State University, 119992 Moscow, Russia; 6Engelhardt Institute of Molecular Biology Russian Academy of Sciences, 119991 Moscow, Russia; chumakovpm@yahoo.com; 7Department of Pathology, Kazan State Medical University, 420012 Kazan, Russia; boichuksergei@mail.ru; 8Biomarker Research Laboratory, Institute of Fundamental Medicine and Biology, Kazan Federal University, 420008 Kazan, Russia; 9Division of Medical and Biological Sciences, Tatarstan Academy of Sciences, 420111 Kazan, Russia

**Keywords:** tumor progression, TP53, cytoskeleton, actin isoforms

## Abstract

**Simple Summary:**

Tumor suppressor p53 dysfunction is one of the most common alterations during tumor progression. Mutations of the TP53 gene occur in more than half of all human neoplasms. Non-muscle cytoplasmic isoforms of actin, beta and gamma, are expressed in all types of eukaryotic cells and play an important role in maintaining cellular architecture, adhesion, contractility, migration, polarization, mitosis, and meiosis. The acquisition of a more aggressive metastatic phenotype by neoplastic cells is associated with changes in the cytoskeleton. Despite the fact that p53 is involved in the regulation of many cell signaling pathways, including regarding cell morphology, motility, and invasion ability, in the presented work, we demonstrate a new universal mechanism for tumor cells acquiring a more metastatic phenotype. The d iscovered molecular mechanism directly links dysfunction of p53 with a shift in the balance of cytoplasmic actins towards the predominance of the gamma isoform, at least in an ERK-dependent manner.

**Abstract:**

Cancer cell aggressiveness, marked by actin cytoskeleton reconfiguration critical for metastasis, may result from an imbalanced ratio favoring γ-actin. Dysfunctional p53 emerges as a key regulator of invasiveness and migration in various cancer cells, both in vitro and in vivo. P53 inactivation (via mutants R175H, R248W, R273H, or TP53 repression) significantly enhanced the migration, invasion, and proliferation of human lung adenocarcinoma A549 cells in vitro and in vivo, facilitating intrapulmonary xenograft metastasis in athymic mice. Conversely, wild-type TP53 (TP53 WT) overexpression in p53-deficient non-small- cell lung cancer (NSCLC) H1299 cells substantially reduced proliferation and migration in vitro, effectively curbing orthotopic tumorigenicity and impeding in vivo metastasis. These alterations in cell motility were closely associated with actin cytoskeleton restructuring, favoring γ-actin, and coincided with ERK1/2-mediated signaling activation, unveiling an innovative regulatory mechanism in malignancy progression. Cancer cell aggressiveness, driven by actin cytoskeleton reorganization and a shift towards γ-actin predominance, may be regulated by p53 dysfunction, thereby providing novel insight into tumor progression mechanisms.

## 1. Introduction

Tumor suppressor p53 is a known regulator of cancer progression. In many human cancers, p53 dysfunction, often caused by TP53 mutations or signaling pathway disruptions, is observed [1]. These mutations frequently cluster within the DNA-binding domain, such as codons 175, 248, 249, 273, and 282, leading to missense mutations that generate stable but non-functional p53 proteins. These mutants fall into two categories: structural mutants affecting the domain’s global structure (e.g., R175H) and contact mutants altering critical DNA-binding amino acids (e.g., R248W and R273H). Many mutant p53 proteins (mutp53) exhibit both dominant-negative activity and gain of function, acquiring oncogenic properties that influence cell cycle, apoptosis, invasion, survival, and metastasis [2,3,4].

Mounting evidence underscores the pivotal roles of p53 and its mutants in regulating cell migration and invasion during cancer metastasis. Mediators affected by wild-type p53 (wtp53) or mutp53 encompass proteins involved in epithelial-to-mesenchymal transition (e.g., SLUG, E-cadherin, and Twist) [5,6], cell–extracellular matrix interactions (e.g., Fibronectin, MT1-MMP, MMP2, and integrins) [7,8,9], growth factors/receptors (e.g., EGFR, PDGF, and TGF-β) [10,11], signaling molecules (e.g., Src, PTEN, and STAT3), Rho GTPases, actin polymerization regulators (e.g., Cdc42, RhoA, and Rac) [12,13], and focal adhesion-related proteins (e.g., Integrins and FAK) [14,15,16].

However, the involvement of p53 and related dysfunctions in actin cytoskeleton remodeling, a key aspect of neoplastic cell transformation, remains underexplored. Additionally, little is known about whether p53 regulates actin expression [16].

Actin, crucial for cell structure and mechanics, comes in six vertebrate isoforms. In non-muscle cells, β-cytoplasmic and γ-cytoplasmic actins (β-actin and γ-actin) are prominent. These cytoplasmic actins play vital roles in cell adhesion, migration, polarization, and cytokinesis. β-actin contributes to adhesion and contraction, while γ-actin’s localization varies with cellular activity, playing a role in cell motility, especially in moving cells, and tight junctions in polarized epithelial cells. Cell transformation chiefly involves actin cytoskeleton reorganization, heightening cell motility and invasion. Cytoplasmic actins assume different roles in neoplastic transformation, with transformed cells typically losing β-actin-containing stress fibrils but displaying well-developed γ-actin-containing networks [17,18,19].

Our recent findings suggest that β-actin acts as a tumor suppressor, inducing epithelial differentiation, inhibiting cell growth, and curbing invasion in lung and colon carcinoma cell cultures, as well as inhibiting tumor growth in vivo. Conversely, γ-actin enhances epithelial tumor cells’ malignancy. The β-/γ-actin balance might serve as an oncogenic marker for carcinomas [19]. However, the mechanisms governing this shift remain insufficiently understood.

In this study, we investigated neoplastic cell properties resulting from p53 tumor suppressor dysfunction. We examined cytoskeletal changes, mobility, in vitro and in vivo growth rates, and metastasis and aimed to identify the mechanisms behind these alterations. We hypothesized that p53 inactivation in carcinoma cells could lead to actin cytoskeleton reorganization by shifting the β-/γ-actin ratio towards γ-actin, thereby stimulating tumor progression and metastasis.

## 2. Materials and Methods

### 2.1. Cell Lines

We employed the following human carcinoma cell lines: HCT116 human colon carcinoma cell line (#CCL-247, ATCC, Manassas, VA, USA), A549 human lung carcinoma cell line (#CCL-185, ATCC, Manassas, VA, USA), and H1299 human lung carcinoma cell line (#CRL-5803, ATCC, Manassas, VA, USA).

### 2.2. p53 DNA Constructs

To generate HCT116 and A549 cell cultures expressing exogenous wild-type and mutant p53 proteins (structural mutant R175H, contact mutant R248W, and contact mutant R273H), we employed lentivirus-mediated gene transfer of corresponding pLV constructs using established procedures [20]. ORFs of wt-p53 and its mutant forms were cloned into XbaI/BamHI sites of pLV-CMV lentivirus construct. DNA accuracies were verified by sequencing.

### 2.3. p53 Inhibition by RNA Interference

For the expression of small interfering RNA (siRNA) specific to p53, we utilized the pLSLP lentiviral construct [21]. Complementary hairpin oligonucleotides containing 19-nucleotide regions corresponding to human p53 mRNA (5′-gactccagtggtaatctac-3′), hairpin structure, and polyA signal were cloned into pLSLP vector by BamHI/EcoRI sites followed by sequencing of the insert.

### 2.4. Generation of pLenti6 with β-Actin and γ-Actin

The creation of pLenti6 constructs for β-actin and γ-actin was previously described [19]. In brief, pLenti6/V5 (K495510, Invitrogen, Carlsbad, CA, USA) was circulated, V5-epitope was deleted, and MCS linker was inserted. ORFs of beta and gamma cytoplasmic actin isoforms, kindly provided by Prof. Christine Chaponnier (University of Geneva, Switzerland), were cloned into the pLenti6 vector. DNA sequencing verification was performed by Evrogen (Evrogen JSC, Moscow, Russia).

### 2.5. Infection and Selection of Infected Cell Cultures

For pseudoviruses production, 293FT (R70007, Invitrogen, Carlsbad, CA, USA) cells were used. Lentiviral DNA constructs, pVSV-G (#8454, Addgene, Watertown, NY, USA) and pΔR8.2 (#12263, Addgene, Watertown, NY, USA), and helper plasmids together with Lipofectamine 2000 reagent (11-668-019, Invitrogen, Carlsbad, CA, USA) were used for transfection. Infected cell cultures were selected for 5–6 days in presence of 1 μg/mL puromycin (A1113803, Gibco, Carlsbad, CA, USA) for pLSLP constructs or 5 μg/mL blasticidin (R210-01, Invitrogen, Carlsbad, CA, USA) for pLenti6 constructs, and 10 μg/mL polybrene (TR-1003-G, Merck, Darmstadt, Germany) was added to virus-containing media. Cells were cultured in DMEMhi (11-055-1N, Biological Industries, Kibbutz Beit-Haemek, Israel) medium supplemented with 10% FBS (S181B-500, Biowest, Nuaillé, France) and penicillin/streptomycin (L0022, Biowest, Nuaillé, France).

### 2.6. Cell Growth Rate Assessment

For cell growth rate determination, 5 × 10^4^ A549, 2.5 × 10^4^ HCT116, and 2.5 ×10^4^ H1299 cells were seeded into 6-well plates, and cell counts were performed at 1–2- day intervals using a hemocytometer (three wells per time point). Measurements continued until monolayer formation.

### 2.7. Boyden Chamber Cell Migration Assay

Migration assays were conducted using transwell Matrigel-coated chambers with membranes of 8 μm pore size (BD Biosciences, Franklin Lakes, NJ, USA) following the manufacturer’s instructions. We used 5 × 10^4^ A549 and H1299 cells and 2.5 × 10^4^ HCT116 cells for the migration assay. Migration activity was quantified by blind counting the migrated cells in at least 10 fields per chamber.

### 2.8. Antibodies

Primary antibodies employed for protein analysis included p53-specific DO-7 monoclonal antibody (m7001, Dako, Glostrup, Denmark), β-actin (MCA5775GA, Bio-Rad, Hercules, CA, USA), γ-actin (MCA5776GA, Bio-Rad, Hercules, CA, USA), p-p44/42 (T202/Y204) (4370S, Cell Signaling, Danvers, MA, USA), ERK1/2 (4695S, Cell Signaling, Danvers, USA), and Pan-actin (MAB1501, Merck, Darmstadt, Germany). AlexaFluor488- and AlexaFluor594-conjugated secondary antibodies (goat anti-mouse and goat anti-rabbit) from Invitrogen, Waltham, MA, USA, were used for immunoblotting.

### 2.9. Western Blot Analysis

Whole-cell extracts were prepared in ice-cold Lysis-M Reagent (4719964001, Roche, Basel, Switzerland) containing protease (04693116001, Roche, Basel, Switzerland) and phosphatase (04906837001, Roche) inhibitors. After a 15 min incubation on ice, the sample was centrifuged at 13,000 rpm for 20 min at +4 °C. Protein concentration in the extracts was determined using a protein assay system (5000205, Bio-Rad, Hercules, CA, USA). An appropriate quantity of protein was separated on an 8–10% SDS polyacrylamide gel and transferred to an Immobilon-FL PVDF Membrane (IPFL00010, Merk, Darmstadt, Germany). Membranes were incubated with primary mouse anti- p53, β-actin, γ-actin, and pan-actin; rabbit anti- ERK1/2, p-ERK1/2, and secondary cross-absorbed Alexa488-conjugated goat anti-mouse (A-11001, Invitrogen, Carlsbad, CA, USA), and anti-rabbit (A-11008, Invitrogen, Carlsbad, CA, USA) antibodies. Typhoon9410 variable mode imager (GE Healthcare, Chicago, IL, USA) was used for band detection. TotalLab v.2.01 software (Newcastle upon Tyne, UK) was used for quantitation of protein bands. Cryopreserved xenograft samples were mechanically homogenized in Lysis-M buffer, and protein extracts were prepared by routine protocol as for cell cultures as listed above.

### 2.10. Immunofluorescent Microscopy

Cells on coverslips were fixed in 2% PFA for 10 min and treated with MeOH at −20 °C for 5 min. Cells were incubated with primary mouse monoclonal antibodies (p53, β-actin, and γ-actin) and secondary AlexaFluor594- or Alexa488-conjugated goat anti-mouse IgG, IgG1, IgG2b antibodies (Jackson ImmunoResearch Laboratories. Inc., West Grove, USA). Nuclear staining was performed using DAPI (Life Technologies, Carlsbad, CA, USA). Images were acquired using a fluorescent Axioplan 2 microscope with 100×/1.3 Plan-Neofluar lens and AxioVision Rel. 4.6 (Carl Zeiss Imaging Systems, Oberkochen, Germany) software.

### 2.11. Nude Mice Assay

For subcutaneous inoculation, female BALB/c nude mice received 10^6^ cells injected into the dorsal flank. Tumor sizes were monitored every 3 days, and tumor volumes were calculated as (width^2^) × (length) × 0.5. After 3–5 weeks, explanted tumors were analyzed. Intrapulmonary inoculation with 10^6^ cells suspended in 20 µL of saline was carried out across the chest’s right side, at the apex point of the costoclavicular perpendicular curve, as previously described [22]. After 5–6 weeks, lungs were removed and analyzed, and 7–10 mice were used perexperimental group, and each in vivo experiment was conducted three times.

### 2.12. Lung Metastases Count

Counting of lung metastases was conducted on serial H&E stained sections throughout the whole lung at 100× magnification. Metastases were counted in all lung lobes except for the middle lobe where the primary tumor was located.

### 2.13. Fluorescent Immunohistochemistry (IHC) Staining

Tissue samples from animals were rinsed in ice-cold PBS, fixed for 24 h in 4% formaldehyde, and embedded in paraffin. Serial 5 μm sections were deparaffinized before immunofluorescent staining. Antigen retrieval was achieved by heating at 95 °C in a target retrieval solution (pH 6.0) (S1699, Dako, Glostrup, Denmark) for 40 min. Sections were then incubated with primary β-/γ-actin antibodies at room temperature for 1 h, followed by incubation with AlexaFluor488- and AlexaFluor594-conjugated secondary antibodies for 30 min at room temperature. Fields of vision were chosen randomly, avoiding edge and necrotic areas. Quantification of immunofluorescence intensity was performed by measuring the specific color intensity using ImageJ (FIJI/ImageJ, version v1.53u) software. For each sample, 5 sections minimum were examined, and at least 15 fields of vision in each section were analyzed.

### 2.14. Statistical Analysis

Statistical analysis was conducted using unpaired Student’s *t*-tests, and data are presented as m ean ± standard error of the mean (SEM) as indicated in figure legends. *p*-values (*) < 0.05 and (**) < 0.01 were considered significant. All experiments were replicated at least three times.

## 3. Results

### 3.1. Impact of p53 Status on Carcinoma Cell Growth and Invasive Behavior In Vitro and In Vivo

To elucidate the roles of p53 in carcinoma cell proliferation and migration, we employed human colon carcinoma HCT116 and lung carcinoma A549 cells, each harboring two endogenous wild-type p53 alleles. Subsequently, we generated HCT116 and A549 sublines with diverse p53 alterations (Figure 1a). The lentiviral delivery of mutant p53 (R175H, R248W, and R273H) and wild-type p53 cDNAs into both HCT116 and A549 cells resulted in a significant elevation of p53 protein levels in all the derived sublines. Conversely, sublines subjected to TP53 repression via shRNA exhibited a substantial reduction in protein levels compared to controls.

Surprisingly, the introduction of p53 mutants, TP53 repression, or exogenous WTp53 expression in HCT116 cells had minimal impact on in vitro growth kinetics (Figure 1b). In A549 cells, exogenous expression of p53 mutants and p53 downregulation induced a slight but statistically significant increase in growth rate. Notably, the introduced WTp53 did not significantly affect the growth rate compared to control cells.

In vitro assays employing the Boyden chamber with M atrigel-coated filters revealed that the expression of p53 mutants or the loss of p53 expression correlated with increased migration and invasive activities (Figure 1c). Conversely, cells expressing exogenous WTp53 displayed a statistically significant, albeit modest, reduction in migration and invasive behavior.

Subsequently, we sought to investigate the in vivo growth and migration of cells with varying p53 alterations.

### 3.2. Influence of p53 Status on Carcinoma Cell Growth In Vivo and β-/γ-Actin Ratio in Xenografts

The expression of mutp53, as well as p53 repression, led to augmented growth of HCT116 and A549 xenografts in athymic mice compared to controls. Notably, the introduction of exogenous WTp53 significantly decelerated HCT116 xenograft growth, while no discernible changes in A549 xenograft growth were observed (Figure 2a).

Given the alterations in migration activity and in vitro growth rates associated with different p53 statuses, we hypothesized potential cytoskeletal changes. Fluorescent IHC staining of xenografts for cytoplasmic actin isoforms revealed shifts in the β-/γ-actin ratio. Specifically, mutp53 expression and p53 repression were associated with a reduction in β-actin content and an increase in γ-actin content compared to control xenografts in both A549 and HCT116 models (Figure 2b and Appendix A). Conversely, xenografts with exogenous WTp53 exhibited a shift in the actin isoform ratio favoring β-actin.

This observed shift in actin isoforms aligns with migration activity; an increased proportion of γ-actin in cells with p53 dysfunction coincides with enhanced cell migration. Consequently, we embarked on investigating the impact of p53 status on metastasis.

### 3.3. Impact of p53 Status on Metastatic Potential in Orthotopic A549 Xenografts

In our study employing an orthotopic intrapulmonary xenograft model of A549 cells, we observed noteworthy effects of p53 status on metastatic activity. Introduction of exogenous mutp53, as well as p53 downregulation, coincided with an approximately twofold increase in the number of lung metastases. Conversely, exogenous WTp53 expression led to an expected twofold reduction in the number of metastases (Figure 3).

To further investigate the potential influence of p53 on malignant characteristics of tumor cells, we employed p53-negative lung carcinoma H1299 cells and a subline featuring exogenous-restored WTp53 expression.

### 3.4. Impact of Exogenous WTp53 on Malignant Traits of H1299 Cells

Utilizing the H1299 cell line featuring the deletion of both TP53 alleles via lentiviral transfer, we successfully generated a subline with restored WTp53 expression (Figure 4a). Introduction of exogenous wild-type TP53 expression resulted in a significant reduction in in vitro migration (Figure 4b), a near-complete suppression of proliferation (Figure 4c), diminished subcutaneous (Figure 4e) and intrapulmonary xenograft size, reduced tumorigenicity, and mitigated metastasis in athymic mice (Figure 4d).

Subsequently, our focus shifted towards investigating the molecular mechanisms underpinning p53-induced changes in neoplastic cells in lung cancer models A549 and H1299.

### 3.5. Impact of p53 Status on Actin Cytoskeleton Remodeling

Given the known influence of the cortical actin cytoskeleton on tumor cell motility, and considering the observed changes in cell proliferation and migration associated with different p53 statuses, we conducted an in-depth investigation of the actin cytoskeleton in A549 and H1299 cell derivatives.

In both A549 and H1299 cell cultures, β-actin staining was moderate to low, characterized by diffuse or disorganized β-actin bundles and a consistently moderate level of cortical γ-actin staining (Figure 5a and Figure 6a). Exogenous mutp53 expression and p53 downregulation in A549 cells induced a more mesenchymal cell phenotype, marked by enhanced γ-actin and reduced β-actin. Morphological changes induced by mutants R175H and R273H resembled those of mutant R248W (as shown in the figure). Introduction of WTp53 inhibited the spread and motile phenotype of both A549 and H1299 cell lines, leading to a more epithelial phenotype characterized by the formation of epithelial islets, achieved by increasing β-actin.

Western blotting confirmed changes in the β-/γ-actin ratio, with a 2–3-fold increase in γ-actin in A549 cells expressing mutp53 or undergoing p53 downregulation (Figure 5b). Expression of WTp53 in both A549 and H1299 cells caused a shift towards β-actin, with a 2–2.5- fold increase (Figure 5b and Figure 6b). This compensatory mechanism for cytoplasmic actin isoform expression was consistently observed in both cell lines. β-a ctin depletion correlated with up regulation of γ-actin and vice versa. This shift towards γ-actin in lung carcinoma cells was associated with a more transformed cellular phenotype. shRNA induced down regulation of p53 exogenous expression constructs in A549 cells with no additional effects on actins’ balance and migration activity (Appendix A). So, there are no data indicating that p53 mutants have any gain-of-function behaviors related to actins’ balance that differ from other p53 inactivation mechanisms.

Furthermore, we examined the impact of p53 status on the intensity of growth factor-stimulated MAPK pathway signaling, a key regulator of cell proliferation and invasion. Western blot analysis revealed that exogenous expression of p53 mutants and p53 downregulation in A549 cells led to ERK1/2 activation, while WTp53 expression inhibited phosphorylation of ERK1/2. Similar results were observed in HCT116 cells, indicating a universal interaction mechanism (Appendix A).

The correlation between ERK1/2 activation and p53-induced changes in actin isoform balance suggests a crucial role for p53 in regulating cytoskeleton dynamics, ultimately influencing cell motility and invasion. This study provides evidence that the observed motility changes are specifically driven by the p53-dependent shift in actin isoform balance and cytoskeletal reorganization rather than other potential effects of p53.

### 3.6. Impact of β- and γ-Actin Overexpression on Malignant Traits and ERK1/2 Signaling

To substantiate the direct influence of actin isoforms on cell migratory and metastatic behavior, we established a series of sublines with overexpressed β- and γ-actin, utilizing a p53-negative H1299 lung carcinoma cell line (Figure 7a).

Elevated β-actin levels resulted in a twofold reduction in cell migration and a slight decrease in proliferation, while γ-actin overexpression significantly increased both migration and proliferation rates (Figure 7b,c).

Intriguingly, β-actin overexpression led to deceleration growth of H1299 xenografts, a 1.6-fold decrease in tumorigenicity, and a notable reduction in metastatic activity when H1299 cells were intrapulmonarily inoculated into athymic mice. Conversely, γ-actin overexpression had no impact on tumorigenicity but augmented growth of xenografts in athymic mice and slightly increased metastatic activity compared to the control group (Figure 7d,e).

Western blot analysis confirmed the shift in the actin isoform ratio in cells with β- or γ-actin overexpression. Specifically, β-actin overexpression inhibited the phosphorylation of ERK1/2, while γ-actin overexpression resulted in the activation of ERK1/2 (Figure 7f).

Thus, the alteration in actin isoform balance, whether induced by β- or γ-actin overexpression, mirrored the effects of changes in p53 status. This suggests that the shift in cell motility due to variations in the actin isoform ratio may indeed occur through a p53-dependent mechanism.

## 4. Discussion

In a previous study, we established the presence of a dynamic equilibrium between β-cytoplasmic and γ-cytoplasmic actin isoforms, where a change in the quantity of one isoform is compensated by a reciprocal alteration in the other. These isoforms exert distinct roles in neoplastic cell transformation, with β-actin acting as a tumor suppressor, influencing epithelial differentiation, cell growth, invasion, and tumor growth in colon and lung carcinoma cells. In contrast, γ-actin amplifies malignant traits and functions as an oncogene [19].

We observed that the prevalence of γ-actin in neoplastic cells triggers ERK1/2 activation, which is corroborated by the co-localization of ERK1/2 and γ-actin in lung carcinoma cells [19]. Considering the widespread activation of ERK1/2 MAP kinases in response to various oncogenic stimuli in most human malignancies, potentially leading to an increase in γ-actin levels, we sought to elucidate previously unknown mechanisms governing the shift in actin isoform ratios. Given the prevalence of p53 alterations in human malignancies, we investigated whether p53 influences the balance of β-/γ-actin in human carcinoma cells.

To explore the effects of various p53 alterations on malignant properties, we established a panel of cell sublines representing common clinical scenarios in colon and lung cancers. We investigated human colon and lung carcinoma cells (HCT116 and A549, respectively) with different p53 statuses: WTp53, exogenous oncogenic p53 hotspot mutant proteins (mutp53), loss of p53 expression, and WTp53 overexpression.

We discovered that exogenous expression of p53 mutant proteins and p53 repression minimally affected cell growth in vitro for HCT116 cells and had only slight effects on A549 cell growth. However, these alterations significantly increased the rate of growth in vivo. Interestingly, these results may be attributed to the impact of mutp53 on tumor neoangiogenesis, as previously observed [20]. We found that xenografts with different p53 statuses displayed changes in the β-/γ-actin isoform ratio. Consequently, p53 dysfunction led to an increase in γ-actin, a hallmark of cell transformation and tumor progression, while diminishing β-actin. Conversely, p53 overexpression resulted in the opposite effect.

The role of p53 in regulating cell motility is well-established and involves various signaling pathways and mechanisms [14,23,24,25]. Loss of p53 is linked to an epithelial– mesenchymal transition (EMT)-like phenotype and the activation of Rho GTPases and associated signaling pathways, which aligns with the increased migration observed in HCT116 and A549 cells following mutp53 introduction or p53 repression.

Cell motility is assessed across different contexts, including 2D surface movement, invasive-type migration through 3D matrices, and metastasis in xenografts [26,27]. While the orthotopic metastasis model for HCT116 human colon cancer cells is challenging [28], we focused on the orthotopic lung cancer cell metastasis model. Our findings revealed that both mutp53 expression and p53 downregulation led to increased invasive-type migration and metastasis of A549 cells, while WTp53 overexpression led to decreased metastasis. A similar trend was observed in p53-negative human lung carcinoma H1299 cells, where exogenous WTp53 restoration suppressed proliferation, invasion, metastasis, and tumorigenicity.

The accelerated growth and metastasis in A549 and H1299 cells with p53 dysfunction coincided with a shift in actin isoform balance towards γ-actin, ERK1/2 activation, and the emergence of a more transformed cellular phenotype.

Direct manipulation of the actin isoform ratio in H1299 cells mirrored these changes in motility and metastasis, as observed in cells with different p53 statuses. A shift towards γ-actin resulted in actin cytoskeleton reorganization, increased motility, and metastasis, as well as induced ERK1/2 activation, akin to cells with p53 dysfunction. Conversely, β-actin predominance led to opposing effects, including the formation of an epithelial phenotype, reduced metastasis, and inhibited ERK1/2 signaling, aligning with cells featuring WTp53. This shift in the actin isoform ratio, favoring γ-actin, heightened cellular motility and likely occurs via a p53-dependent mechanism. Therefore, elevated γ-actin levels represent a fundamental characteristic of cell transformation and tumor progression, with a shift in actin isoform ratios being a universal feature in the development of prevalent tumors (Figure 8).

Notably, γ-actin’s ability to interact with and modulate the activity of signaling molecules, such as PP1α and ERK1/2, along with its role in actin assembly via interaction with key regulatory proteins of the actin cytoskeleton (e.g., p34-Arc, WAVE2, and cofilin1), position it as an oncogene implicated in tumorigenesis [19,29]. P53-dependent influence on ERK signaling is well- understood. Based on this fact and our own published data [19] concerning direct actin– ERK interaction, the result of our work can be considered the identification of a new type of “P53-ERK-actins” signaling. This suggests that restoring and maintaining a balanced ratio of non-muscle actin isoforms by reducing γ-actin expression could open new avenues for enhancing anticancer therapy [30].

In the current clinical landscape, cytoskeletal-targeting chemotherapeutic drugs are utilized for treating specific neoplasms [31,32]. These drugs primarily affect the tubulin components of the cytoskeleton, influencing depolymerization or prevention of polymerization (e.g., vinblastine [33]) and microtubule stabilization (e.g., paclitaxel [34,35]), thus impeding cancer cell growth. However, in contrast to microtubules, actin-associated drugs present a more intricate challenge. The actin cytoskeleton and its associated proteins constitute a complex target in anticancer therapy, primarily due to their involvement in various cellular processes and the frequent occurrence of cardiotoxic side effects resulting from actin and actin-binding protein (ABP) regulation [36,37]. Moreover, inhibitors that target actin exhibit substantial effects on cell shape, migration, division, and other actin-based processes in animal cells by impacting actin cytoskeleton function and organization. These inhibitors lack specificity for actin isoforms, leading to unacceptable off-target effects.

The advent of specific monoclonal antibodies enabling the discrimination of cytoplasmic actin isoforms has facilitated the study of β- and γ-cytoplasmic actin distribution in cells and the impact of available inhibitors on β- and γ-actin-containing structures. It has been revealed that these inhibitors exert distinct effects on β- and γ-actins. Given that the cellular ratio of β-/γ-actins can fluctuate, it is imperative to consider their differential impact on actin isoforms when selecting inhibitors to achieve the desired therapeutic effects.

## 5. Conclusions

The findings reveal that p53 dysfunction plays a pivotal role in regulating cancer cell invasiveness and migration by impacting actin cytoskeleton reorganization and actin isoform balance, shedding light on a novel tumor progression mechanism.

## Figures and Tables

**Figure 1 cancers-16-01123-f001:**
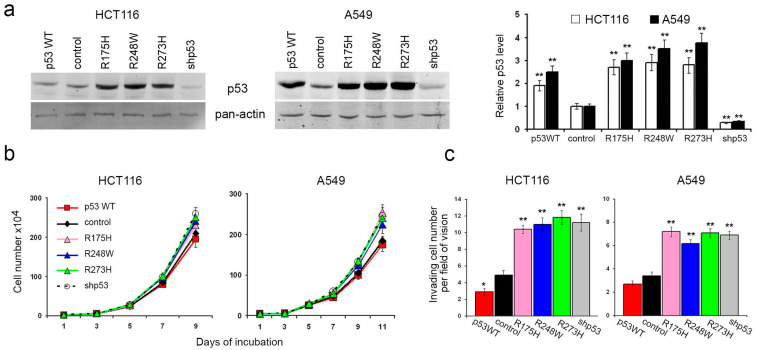
Effects of exogenous WTp53, mutp53, and p53 repression on proliferation and motility of HCT116 and A549 cells in vitro: p53 dysfunction stimulates invasion and has no effect on proliferation. (**a**). Western blot (WB) analysis of HCT116 and A549 cells with exogenous p53WT, mutp53, and p53 repression. Representative images are shown. Graphs represent the relative p53 level compared to control cells (mean ± SEM). (**b**). Proliferation dynamics of HCT116 and A549 cells with various p53 statuses. Error bars represent SEM. (**c**). Invasion activity of HCT116 and A549 cells with various p53 statuses through Matrigel-coated membranes. Graphs represent mean ± SEM. Uncropped Western Blots can be found at Appendix A.

**Figure 2 cancers-16-01123-f002:**
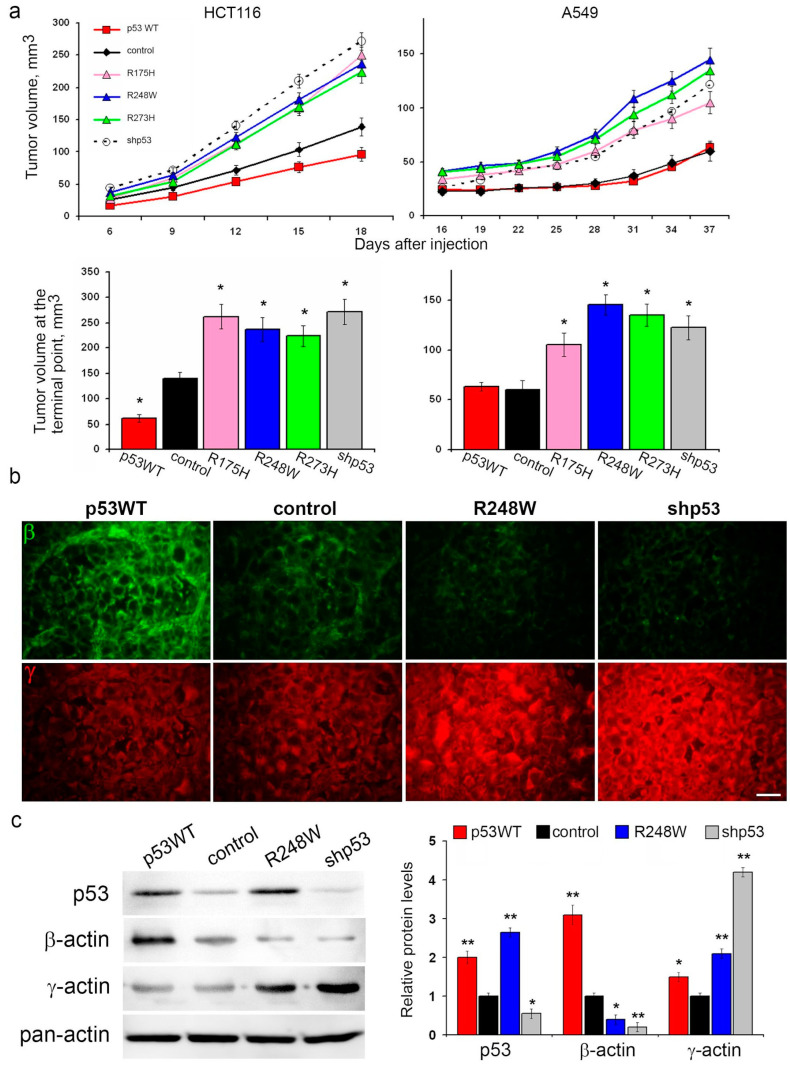
Effects of p53 status on malignant growth in vivo: p53 dysfunction stimulates xenograft growth and leads to an increase in γ-actin content compared to control xenografts. (**a**). The growth rate of subcutaneous xenografts after injection of HCT116 (left) and A549 (right) cells with exogenous WTp53, mutant proteins, and silenced p53. Error bars represent SEM. The lower panel shows the tumor volume at the terminal measurement point. Graphs represent mean ± SEM. (**b**). Fluorescent immunohistochemistry (IHC) staining for β-actin (green) and γ-actin (red) of A549 subcutaneous xenografts with various p53 statuses. The scale bar represents 50 µm. (**c**). WB analysis of A549 subcutaneous xenografts with various p53 statuses 35 days after injection. Representative images are shown. Graphs represent relative p53 and β-/γ-actin levels compared to control (m ean ± SEM). Uncropped Western Blots can be found at Appendix A.

**Figure 3 cancers-16-01123-f003:**
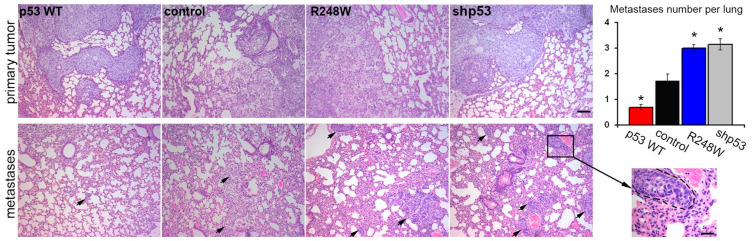
Effects of p53 status on growth and metastasis of intrapulmonary xenograft A549 in athymic mice: p53 dysfunction stimulates experimental metastasis. H & E staining of lung sections 4–5 weeks after intrapulmonary injection: upper panel—primary tumor localized in the middle lobe of the right lung; lower panel—metastases in other lobes of the lung are indicated with arrows. The whole lung was sectioned and analyzed. The scale bar represents 100 µm for crop image—25 µm. Graphs represent the number of metastases per lung (mean ± SEM).

**Figure 4 cancers-16-01123-f004:**
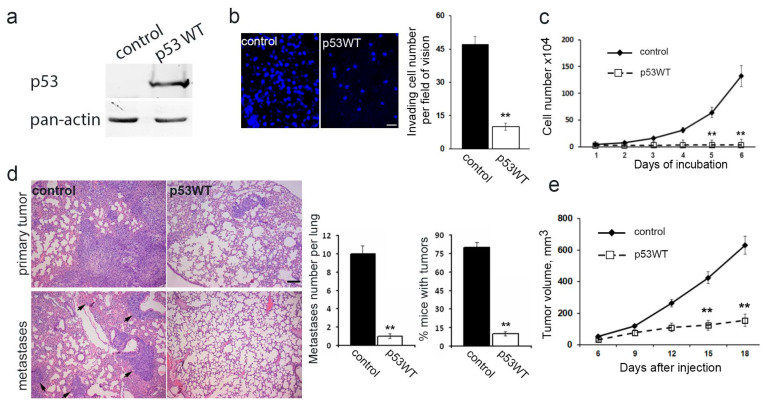
Effects of exogenous WTp53 in H1299 cells on motility, proliferation, growth, and metastasis formation: restored WTp53 suppresses malignant features of neoplastic cells. (**a**). Exogenous-restored WTp53 expression in p53-negative H1299 cells. Typical images are presented. (**b**). Invasive activity of H1299 with exogenous WTp53 through Matrigel-coated membranes. Typical fields of vision are presented, DAPI staining, and scale bar represents 50 µm (**left**). Graphs represent mean ± SEM (**right**). (**c**). H1299 proliferation dynamics with exogenous WTp53. Error bars represent SEM. (**d**). Growth and metastasis of intrapulmonary xenograft H1299 cells with exogenous WTp53. The whole lung was sectioned and analyzed. H&E staining of lung sections 4 weeks after intrapulmonary injection: upper panel—primary tumor localized in the middle lobe of the right lung; lower panel—metastases in other lung lobes are indicated with arrows. The scale bar represents 100 µm (**left**). Graphs represent the number of metastases per lung and % mice with intrapulmonary xenografts (m ean ± SEM) (**right**). (**e**). The growth rate of subcutaneous xenografts after injection of H1299 cells with exogenous WTp53. Error bars represent SEM. Uncropped Western Blots can be found at Appendix A.

**Figure 5 cancers-16-01123-f005:**
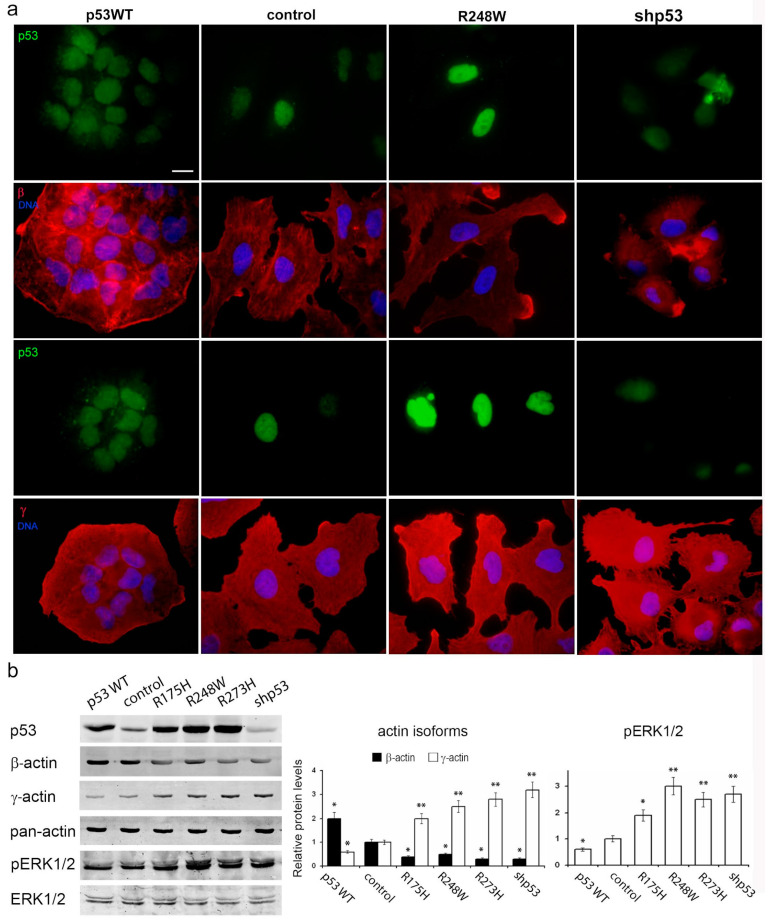
Effects of p53 status on actin cytoskeleton reorganization and shift in the actin isoforms ratio in A549 cells: p53 dysfunction induces formation of a mesenchymal cell phenotype, marked by enhanced γ-actin and reduced β-actin, as well as pERK1/2 activation. (**a**). Immunofluorescent staining for β-actin, γ-actin, and p53 of A549 cells with altered p53 expression. Scale bars represent 10 µm. (**b**). WB analysis of A549 cells with various p53 statuses. Representative images are shown. Graphs represent relative β-/γ-actin and pERK1/2 levels compared to control cells (m ean ± SEM). Uncropped Western Blots can be found at Appendix A.

**Figure 6 cancers-16-01123-f006:**
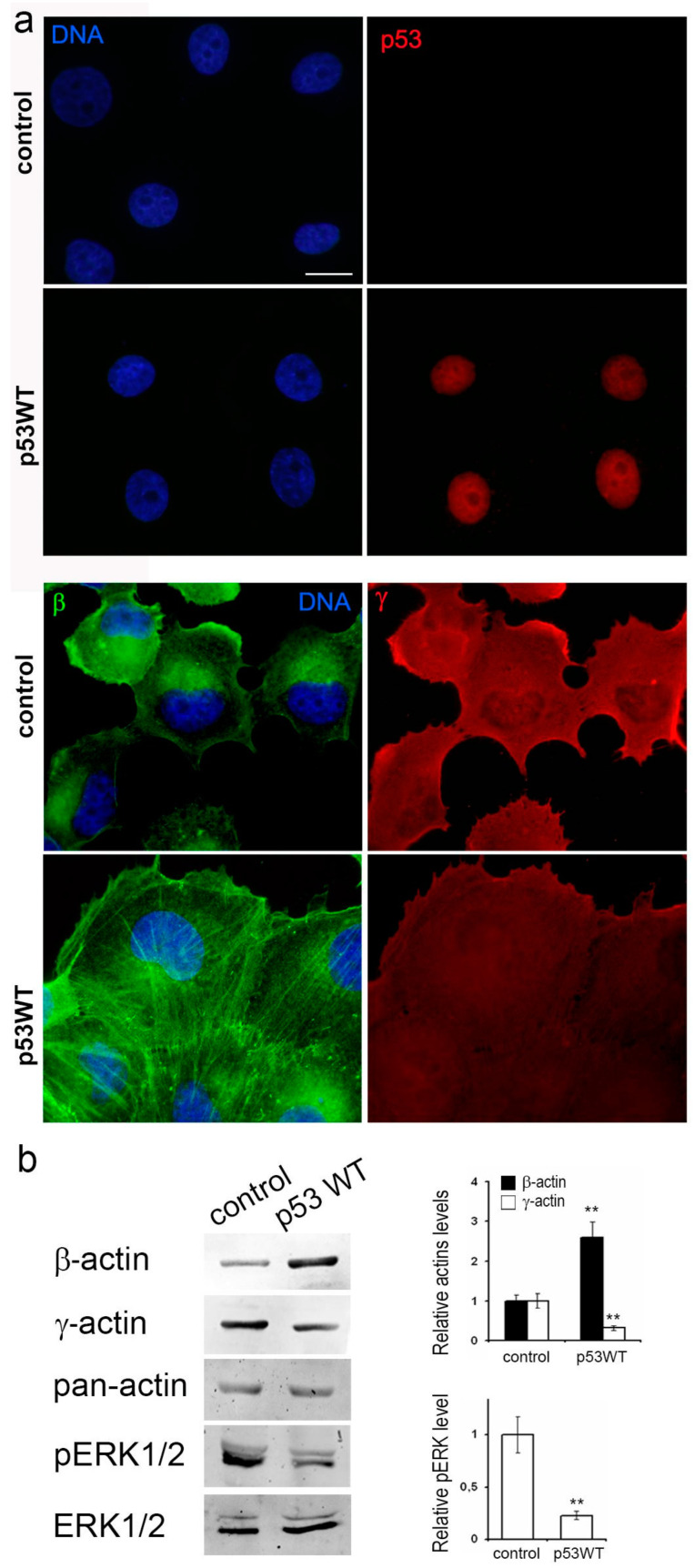
Effects of p53 status on actin cytoskeleton reorganization and shift in the actin isoforms ratio in H1299: WTp53 induces formation of a more epithelial phenotype, achieved by increasing β-actin and ERK1/2 suppression. (**a**). Immunofluorescent staining for β-actin, γ-actin, and p53 of H1299 cells with altered p53 expression. Scale bars represent 10 µm. (**b**). WB analysis of H1299 cells with exogenous WTp53. Representative images are shown. Graphs represent relative β-/γ-actin and pERK1/2 levels compared to control cells (mean ± SEM). Uncropped Western Blots can be found at Appendix A.

**Figure 7 cancers-16-01123-f007:**
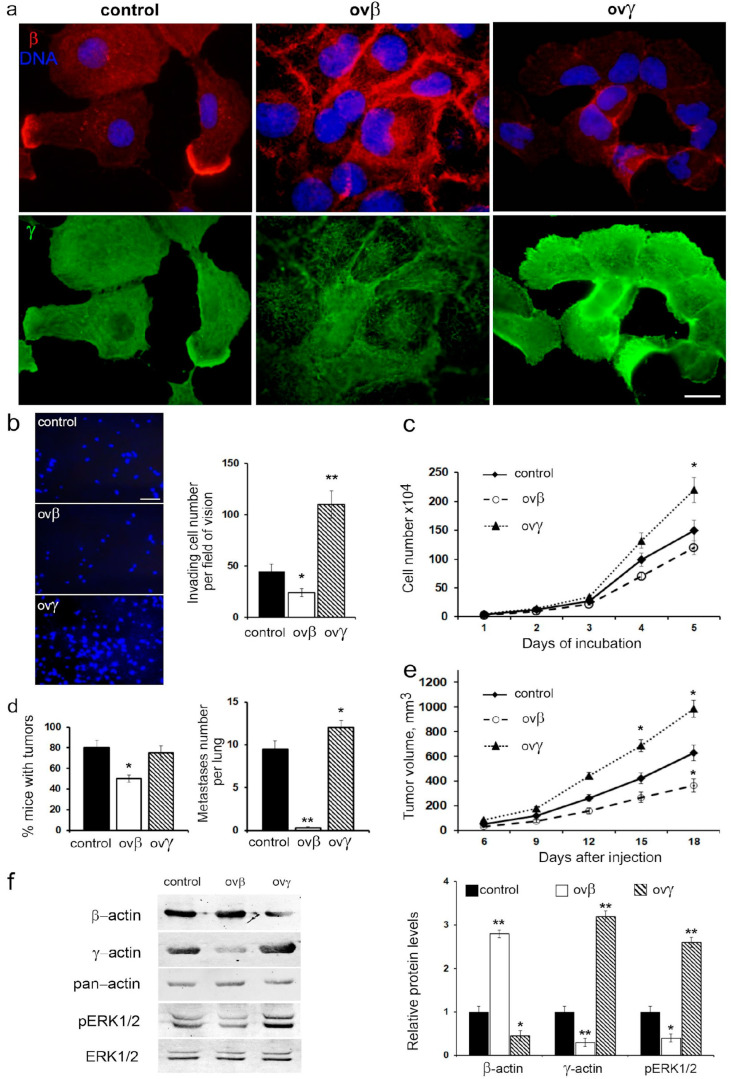
Effects of cytoplasmic actin isoforms on H1299 malignant characteristics and ERK1/2 activation: γ-actin enhances and β-actin, on the contrary, reduces the manifestation of neoplastic cells’ features. (**a**). Immunofluorescent staining for β-actin and γ-actin of H1299 cells. Scale bars represent 10 µm. (**b**). Invasive activity of H1299 subline cells through Matrigel-coated membranes. DAPI staining of invading cells and typical fields of vision are presented. The scale bar represents 50 µm (**left**). Graphs represent mean ± SEM (**right**). (**c**). Proliferation dynamics of H1299 with overexpressed β-actin and γ-actin. Error bars represent SEM. (**d**). Tumorigenicity (% of mice with formed intrapulmonary tumors) (**left**) and number of metastases per lung (**right**) in H1299 cells with exogenously expressed β-actin and γ-actin. Graphs represent mean ± SEM. (**e**). The growth rate of subcutaneous xenografts after injection of H1299 cells with exogenous expression of β- or γ-actins. Error bars represent SEM. (**f**). WB analysis of H1299 cells with exogenous expression of β- or γ-actins. Representative images are shown. Graphs represent relative β-/γ-actin and pERK1/2 levels (mean ± SEM). Uncropped Western Blots can be found at Appendix A.

**Figure 8 cancers-16-01123-f008:**
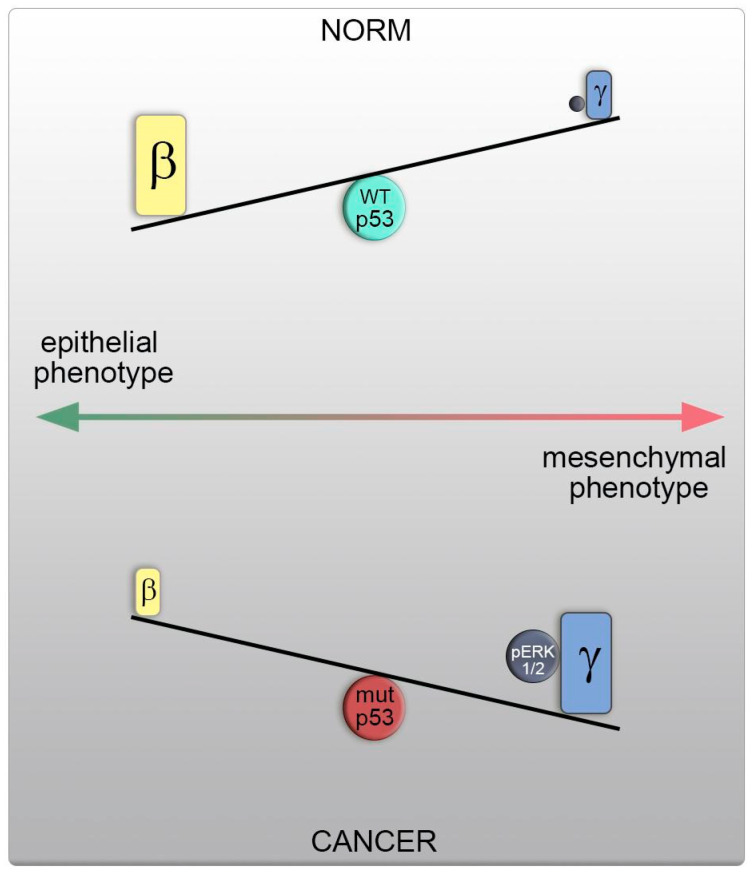
The shift towards γ-actin, a modest oncogene that enhances cellular mobility, is driven by ERK1/2 activation in cells with p53 dysfunction, ultimately amplifying the malignant properties of neoplastic cells.

## Data Availability

The original contributions presented in the study are included in the article/Appendix A; further inquiries can be directed to the corresponding author.

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
