# Peer review of "Actin-Dependent Mechanism of Tumor Progression Induced by a Dysfunction of p53 Tumor Suppressor"

_cancers, 2024, doi:10.3390/cancers16061123_

Round 1

Reviewer 1 Report

Comments and Suggestions for Authors

The premise of the paper was that cancer cell aggressiveness correlating with actin cytoskeleton reconfiguration may result from an imbalanced ratio of actin isoforms favoring γ-actin. p53 inactivation via mutations R175H, R248W, R273H or TP53 repression enhanced migration, invasion, and proliferation of human lung adenocarcinoma A549 cells in vitro and in vivo, facilitating intrapulmonary xenograft metastasis in athymic mice. Conversely, wild-type TP53 overexpression in p53-deficient non-small cell lung cancer H1299 cells reduced proliferation and migration in vitro, effectively reduced orthotopic tumorigenicity and impeded metastasis. These alterations were associated with actin cytoskeleton restructuring favoring γ-actin and with ERK1/2- signaling activation, which appears to be a novel regulatory mechanism in tumor progression. It is concluded that cancer cell aggressiveness driven by actin cytoskeleton reorganization with a shift towards γ-actin may be regulated by p53 dysfunction. The paper is well executed and written, with proper controls and statistics. It has a nice complementation of in vitro and in vivo experiments. Provided raw data show lack of result manipulation.

This reviewer has some minor concerns about the manuscript.

1. In the statistics section please change ≤ to <. This section states how many repeats were made. Please also state how many mice were used per each experiment and how many in vivo experiments were conducted (here or in the figure legends).

2. On Fig. 1 and below, please specify what pL6 is; this term is not found in the Methods. If this is a pLV plasmid transduction only, please explain why it has different p53 expression from WT and what is the p53 status of pL6. The same problem concerns Fig. 2c and below.

3. Was any significant difference observed between curves in Fig. 2a?

4. Figs. 2b, 5a and 7a need revisiting. The panels appear fuzzy and some are oversaturated. This might be due to downsampling by the journal file converter, which may be circumvented by using pdf format of the pictures.

5. On Fig. 3 please consider increasing magnification on the metastasis row.

6. It looks like the pL6 line used in figures 1-3 is different from the line with the same name on Fig. 4. Please reconcile. Also, please increase brightness of Fig. 4b and 7b.

7. Please justify the use of ERK only in the MAPK cascades study. Or list it as a limitation of the study that might have included Akt and/or p38.

Comments on the Quality of English Language

Please correct some unconventional terms (norma to norm, Fig. 8; cryosafed to cryopreserved, section 2.9).

Author Response

Rev1

The premise of the paper was that cancer cell aggressiveness correlating with actin cytoskeleton reconfiguration may result from an imbalanced ratio of actin isoforms favoring γ-actin. p53 inactivation via mutations R175H, R248W, R273H or TP53 repression enhanced migration, invasion, and proliferation of human lung adenocarcinoma A549 cells in vitro and in vivo, facilitating intrapulmonary xenograft metastasis in athymic mice. Conversely, wild-type TP53 overexpression in p53-deficient non-small cell lung cancer H1299 cells reduced proliferation and migration in vitro, effectively reduced orthotopic tumorigenicity and impeded metastasis. These alterations were associated with actin cytoskeleton restructuring favoring γ-actin and with ERK1/2- signaling activation, which appears to be a novel regulatory mechanism in tumor progression. It is concluded that cancer cell aggressiveness driven by actin cytoskeleton reorganization with a shift towards γ-actin may be regulated by p53 dysfunction. The paper is well executed and written, with proper controls and statistics. It has a nice complementation of in vitro and in vivo experiments. Provided raw data show lack of result manipulation.

This reviewer has some minor concerns about the manuscript.

Raw data provided

  1. In the statistics section please change ≤ to <.This section states how many repeats were made.

Please also state how many mice were used per each experiment and how many in vivo experiments were conducted (here or in the figure legends).

 ≤ to < changed

We added this information to Nude mice assay section.

  1. On Fig. 1 and below, please specify what pL6 is; this term is not found in the Methods. If this is a pLV plasmid transduction only, please explain why it has different p53 expression from WT and what is the p53 status of pL6. The same problem concerns Fig. 2c and below.

Term pL6 (pLenti6) is lentivirus control - empty plasmid transducted in corresponding cells. p53 status of  "pl6" differs in various cancer cell lines; wild-type p53 is expressed in HCT116 and A549, H1299 are p53-negative cells. To avoid misunderstanding, we replaced pL6 by control in the figures and their legends.

  1. Was any significant difference observed between curves in Fig. 2a?

Significant difference was observed between curves and was presented at the lower panel in Fig.2a, showing xenograft volumes at the terminal measurement point.

  1. Figs. 2b, 5a and 7a need revisiting. The panels appear fuzzy and some are oversaturated. This might be due to downsampling by the journal file converter, which may be circumvented by using pdf format of the pictures.

Indeed, the quality of the 2B, 5A and 7A pictures is associated with export to Word Format. And we attach all figures in the PDF format, and the downstairs, we can send the original tif files.

  1. On Fig. 3 please consider increasing magnification on the metastasis row.

We added a crop image representing metastasis.

  1. It looks like the pL6 line used in figures 1-3 is different from the line with the same name on Fig. 4. Please reconcile.

In Figures 1-3 and 5 control ‘pL6’ was p53-WT-cells A549 and/or HCT116, in Figures 4, 6, 7 ‘pL6’ was p53-negative cells H1299.

Also, please increase brightness of Fig. 4b and 7b.

Brightness of Fig. 4b and 7b increased.

  1. Please justify the use of ERK only in the MAPK cascades study. Or list it as a limitation of the study that might have included Akt and/or p38.

We are well understanding that P53 is involved in many cellular signaling pathways, but in this work we limited ourselves to the study of the change in the ERK1/2, since earlier (Dugina, V.; Khromova, N.; Rybko, V.; Blizniukov, O.; Shagieva, G.; Chaponnier, C.; Kopnin, B.; Kopnin P. Tumor promotion by γ and suppression by β non-muscle actin isoforms. Oncotarget 2015, 6, 14556-14571.  https://doi.org/10.18632/oncotarget.3989) we showed a direct interaction between gamma actin and ERK1/2 exactly. To avoid possible speculation, we proposed this possible p53-induced and ERK-dependent molecular mechanism of tumor progression based on beta/gamma actin balance changes.

Reviewer 2 Report

Comments and Suggestions for Authors

As an extension of their previous study, this manuscript reported the relationship between p53 dysfunction, actin cytoskeleton reorganization, and tumor progression. It finds that p53 mutations shifted the balance of non-muscle β- and γ-actins towards γ-actin, leading to increased cell motility and invasiveness, both in vitro and in vivo. The experiments are well-designed, and the manuscript is well-written. These results may provide new insights into tumor progression mechanisms. I suggest only minor revisions as follows before its acceptance.

Please describe the detailed experimental protocols directly in the manuscript rather than just referring to earlier publications (Refs 20-22). This includes methods for DNA constructs to engineer HCT116 and A549 cells, RNA interference, and generation of pLenti6 with β-actin and γ-actin.

Refer to Figure 3 in the text of section 3.3.

Author Response

Rev2

As an extension of their previous study, this manuscript reported the relationship between p53 dysfunction, actin cytoskeleton reorganization, and tumor progression. It finds that p53 mutations shifted the balance of non-muscle β- and γ-actins towards γ-actin, leading to increased cell motility and invasiveness, both in vitro and in vivo. The experiments are well-designed, and the manuscript is well-written. These results may provide new insights into tumor progression mechanisms. I suggest only minor revisions as follows before its acceptance.

Please describe the detailed experimental protocols directly in the manuscript rather than just referring to earlier publications (Refs 20-22). This includes methods for DNA constructs to engineer HCT116 and A549 cells, RNA interference, and generation of pLenti6 with β-actin and γ-actin.

Additional technical information was added to the manuscript. Maps of used plasmids can be provided upon request.

Refer to Figure 3 in the text of section 3.3.

Referred.

Reviewer 3 Report

Comments and Suggestions for Authors

This research is exemplary in its design and execution, demonstrating an outstanding level of control and precision. It is underpinned by an impressive array of evidence, complemented by a thorough analysis from diverse viewpoints, which adds significant depth to the study's findings. The manuscript is articulate and informative, with a clear and engaging presentation style. The figures are presented and described with exceptional clarity, facilitating an easy understanding of complex data. I only have a few constructive comments to offer for consideration.

Minor comment:

1.     It’s impossible to distinguish those curves in the black-and-white format in Figure 1b, Figure 2a.

2.     Please add a p53 blot for Figure 5b.

3.     Given the importance of p53 in regulating numerous downstream effectors, I wonder if the author has conducted any assays to specifically restrict the transition from β-actin to γ-actin in scenarios involving either mutant p53 or p53 knockdown. This investigation could help determine if actin serves as a primary effector downstream of p53.

4. Considering the observed correlation between p53 and actin in cell lines, I am curious if the author has explored online databases to investigate whether mutations in p53 correlate with changes in actin expression in patient samples. 

Author Response

Rev3

This research is exemplary in its design and execution, demonstrating an outstanding level of control and precision. It is underpinned by an impressive array of evidence, complemented by a thorough analysis from diverse viewpoints, which adds significant depth to the study's findings. The manuscript is articulate and informative, with a clear and engaging presentation style. The figures are presented and described with exceptional clarity, facilitating an easy understanding of complex data. I only have a few constructive comments to offer for consideration.

Minor comment:

1.It’s impossible to distinguish those curves in the black-and-white format in Figure 1b, Figure 2a.

In Figure 1b and 2a we added color curves. Other histograms of these figures are colored with corresponding colors as well.

  1. Please add a p53 blot for Figure 5b.

P53 blot is added.

  1. Given the importance of p53 in regulating numerous downstream effectors, I wonder if the author has conducted any assays to specifically restrict the transition from β-actin to γ-actin in scenarios involving either mutant p53 or p53 knockdown. This investigation could help determine if actin serves as a primary effector downstream of p53.

This is our first work showing the impact of changes in p53 status on the balance of cytoplasmic actin isoforms. We are well understanding that P53 is involved in many cellular signaling pathways, but in this work we limited ourselves to the study of the change in the ERK1/2, since earlier (Dugina, V.; Khromova, N.; Rybko, V.; Blizniukov, O.; Shagieva, G.; Chaponnier, C.; Kopnin, B.; Kopnin P. Tumor promotion by γ and suppression by β non-muscle actin isoforms. Oncotarget 2015, 6, 14556-14571.  https://doi.org/10.18632/oncotarget.3989) we showed a direct interaction between gamma actin and ERK1/2 exactly. To avoid possible speculation, we proposed this possible p53-induced and ERK-dependent molecular mechanism of tumor progression based on beta/gamma actin balance changes. Considering the versatility of p53 functions, most likely the variant of regulation we proposed is not the only one. Determining the significance of p53- and ERK-dependent actins regulation or/and investigation of alternative signaling pathways of such regulation need additional and huge research.

  1. Considering the observed correlation between p53 and actin in cell lines, I am curious if the author has explored online databases to investigate whether mutations in p53 correlate with changes in actin expression in patient samples.

We have studied and are studying this issue. But unfortunately, such research is difficult to do now due to technical problems. There are no problems with the analysis of p53 in clinical material, but they are present in the analysis of beta and gamma isoforms of actin. This applies to both their protein analysis and mRNA expression. Since their protein molecules differ only in 4 N-terminal amino acids very problematic their discrimination even Mass Spec methods for example. We performed such study and were dissatisfied with its results. In our opinion, the only reliable method for protein detection of beta and gamma actin is the use of isoforms of specific antibodies. But unfortunately, in a huge number of published works and data, when the authors talk about determining primarily beta actin, based on the statements of the manufacturers of antibodies, they actually determine pan-cytoplasmic actin (beta and gamma together), because antibodies are not actually isoform specific. Despite the lower homology in mRNA sequences, problems with quantitative determination of expression using NGS sequencing are still present. We also confirmed this during the transcriptome NGS analysis of our cell lines with altered expression of actin isoforms using shRNA. Perhaps this can be avoided using methods for sequining large fragments, for example using nanopore technology. And since we do not have any confidence in this data now, we did not include such a section at all, despite our interest. The only reliable research option for us seems to be isoform-specific IHC analysis of clinical material with a known p53 status. In principle, we are planning similar studies if the issues of funding, the possibility of obtaining clinical material and permission from the ethics committee are resolved.